# Phenolic Profile of Herbal Infusion and Polyphenol-Rich Extract from Leaves of the Medicinal Plant *Antirhea borbonica*: Toxicity Assay Determination in Zebrafish Embryos and Larvae

**DOI:** 10.3390/molecules25194482

**Published:** 2020-09-29

**Authors:** Bryan Veeren, Batoul Ghaddar, Matthieu Bringart, Shaymaa Khazaal, Marie-Paule Gonthier, Olivier Meilhac, Nicolas Diotel, Jean-Loup Bascands

**Affiliations:** INSERM, UMR 1188, Diabète Athérothrombose Thérapies Réunion Océan Indien (DéTROI), Plateforme CYROI, Université de La Réunion, 2 rue Maxime Rivière, 97490 Sainte-Clotilde, Reunion, France; bryan.veeren@univ-reunion.fr (B.V.); batoul.ghaddar@univ-reunion.fr (B.G.); matthieu.bringart@inserm.fr (M.B.); shaymaa.khazaal@hotmail.com (S.K.); marie-paule.gonthier@univ-reunion.fr (M.-P.G.); olivier.meilhac@inserm.fr (O.M.); nicolas.diotel@univ-reunion.fr (N.D.)

**Keywords:** *Antirhea borbonica*, medicinal plants, polyphenols, zebrafish, toxicity, LC-MS/MS

## Abstract

*Antirhea borbonica* (*A. borbonica*) is an endemic plant from the Mascarene archipelago in the Indian Ocean commonly used in traditional medicine for its health benefits. This study aims (1) at exploring polyphenols profiles from two types of extracts—aqueous (herbal infusion) and acetonic (polyphenol rich) extracts from *A. borbonica* leaves—and (2) at evaluating their potential toxicity in vivo for the first time. We first demonstrated that, whatever type of extraction is used, both extracts displayed significant antioxidant properties and acid phenolic and flavonoid contents. By using selective liquid chromatography–tandem mass spectrometry, we performed polyphenol identification and quantification. Among the 19 identified polyphenols, we reported that the main ones were caffeic acid derivatives and quercetin-3-*O*-rutinoside. Then, we performed a Fish Embryo Acute Toxicity test to assess the toxicity of both extracts following the Organisation for Economic Cooperation and Development (OECD) guidelines. In both zebrafish embryos and larvae, the polyphenols-rich extract obtained by acetonic extraction followed by evaporation and resuspension in water exhibits a higher toxic effect with a median lethal concentration (LC_50_: 5.6 g/L) compared to the aqueous extract (LC_50_: 20.3 g/L). Our data also reveal that at non-lethal concentrations of 2.3 and 7.2 g/L for the polyphenol-rich extract and herbal infusion, respectively, morphological malformations such as spinal curvature, pericardial edema, and developmental delay may occur. In conclusion, our study strongly suggests that the evaluation of the toxicity of medicinal plants should be systematically carried out and considered when studying therapeutic effects on living organisms.

## 1. Introduction

Réunion island, a French volcanic overseas department belonging to the Mascarene Archipelago (Indian Ocean), has never been connected to any other landmasses [1] and is described as one of the 36 world biodiversity hotspots [2]. It displays a wide and rich flora with a high percentage of endemic species. Many of the indigenous and endemic plants from Reunion island have been and are still used for traditional medicine [3]. Although some studies have reported the potential therapeutic effects of these plants in combatting hypertension [4], oxidative stress, inflammation [5], parasitosis (i.e., plasmodium), and viruses (i.e., Chikungunya, Dengue, and Zika) [6,7,8], their deep content characterization as well as their real efficiency in vivo remains largely unknown.

Since 2012, 22 medicinal plants have been registered at the French pharmacopeia [9]. Among these medicinal plants, *Antirhea borbonica* (*A. borbonica*) leaves are peculiarly interesting, as they are widely used in traditional medicine for treating, among other things, diabetes, urinary tract infection, diarrhea, hemorrhage, rheumatism, and also kidney stones [3,10]. Most of these interesting presumptive effects have been attributed to the antioxidant and anti-inflammatory properties of *A. borbonica* leaves. Based on these beneficial effects, it has been previously reported that polyphenol-rich extracts from *A. borbonica* exhibited strong antioxidant and anti-inflammatory effects, in vitro, on preadipocytes, cerebral endothelial cells, and red blood cells [5,11,12], as well as, in vivo, in a mouse stroke model [13] and a diet-induced overweight zebrafish model [14]. Importantly, these antioxidant and anti-inflammatory biological effects were associated with the capacity of polyphenols to regulate key molecular targets, such as ROS-producing and detoxifying enzymes and the redox-sensitive transcriptional factor Nrf2, and improve vasoactive markers in these in vitro and in vivo pathological models [5,11,12,13,14].

Because *A. borbonica* seems to display therapeutic effects correlated to its polyphenol content, a thorough investigation was required to determine its precise phenolic profile composition and its subsequent potential toxicity. To the best of our knowledge, although registered in French pharmacopeia and despite the various therapeutic effects suggested in a number of in vitro studies (see above), no developmental toxicity study has been reported for any of these 22 medicinal plants.

In the present study, we compared the precise phenolic profile of aqueous and acetonic (polyphenol rich) extracts from dried leaves of *A. borbonica* by performing LC-MS/MS analysis. In a second part, we investigated the potential toxicity of several concentrations of these extracts using a zebrafish model. Zebrafish (Danio rerio), due to its small-size, high reproductive ability, and rapid embryogenesis and organogenesis, has become the most famous cost-effective alternative model used for large-scale and high-throughput toxicological and physiopathological studies [15,16]. The transparency of zebrafish embryos and larvae enables the real-time visualization and imaging of drug effects throughout the developmental process. This laboratory model is widely used to test compounds’ toxicity. We consequently determine the median lethal concentration (LC_50_) in a zebrafish embryo and larvae models.

## 2. Results

### 2.1. Determination of Total Antioxidant Activity and Phenolic/Flavonoid Contents of Aqueous and Acetonic Extracts

In their traditional use, *A. borbonica* leaves are used for herbal infusion. We assessed the antioxidant capacity of *A. borbonica* aqueous extract and compared it with a polyphenols-rich extract obtained by acetonic solvent-assisted extraction, which is supposed to contain the maximal yield of polyphenols. To this end, an 2,2-Diphenyl-1-picrylhydrazyl (DPPH) assay was performed. As shown in Figure 1A, both extracts displayed, compared to ascorbic acid (40 g GAE/L) (94.7% ± 0.5%), an important antioxidant capacity of up to 90.7% ± 0.6% and 84.8% ± 1.2% (* *p* < 0.05 vs. 40 g/L of acetonic extract and ^$$^
*p* < 0.01 vs. 40 g/L (GAE) ascorbic acid) for polyphenol rich and aqueous extracts, respectively. The DPPH assay on both *A. borbonica* extracts at different concentrations of 40, 30, 22.5, 16.9, 12.7, 9.5, 7.2, and 2.3 g GAE/L is shown in Appendix A. The antioxidant activity is reported in Table 1 as IC_50_, the required concentration for a 50% reduction in DPPH radicals. There was a minimum IC_50_ value of 3.1 ± 0.3 g/L for the acetonic extract, followed by the IC_50_ value of the aqueous extract at 3.3 ± 0.3 g/L. These results confirm the important free radical-scavenging activity of both extracts compared to ascorbic acid (2.8 ± 0.1 g/L).

The total phenolic acid and flavonoid contents of acetonic and aqueous extracts were measured by the Folin–Ciocalteu and aluminum chloride colorimetric methods, respectively, for the following range of concentrations of *A. borbonica* (40, 30, 22.5, 16.9, 12.7, 9.5, 7.2, and 2.3 g/L) (Appendix A). The highest phenolic content was exhibited by the acetonic extract with 1778.9 ± 34.1 (*** *p* < 0.001 (vs. 40 g/L of acetonic extract) mg GAE/100 g dried powder, followed by the aqueous extract with 1146.9 ± 14.7 mg GAE/100 g dried powder at a concentration of 40 g/L (Figure 1B). As shown in Figure 1C, the acetonic extract exhibited the highest flavonoid content with 1005.6 ± 19.3 mg QE/100 g dried powder, followed by the aqueous extract with 648.3 ± 8 (*** *p* < 0.001 (vs. 40 g/L of acetonic extract) mg QE/100 g dry powder at the concentration of 40 g/L.

### 2.2. Characterization of Polyphenols from Antirhea borbonica Acetonic Extract

In order to determine the composition of *A. borbonica* acetonic extract, a high-resolution accurate mass spectrometry analysis was performed using a Q-Exactive™ Plus mass spectrometer (Table 2). The identification of polyphenols was based on their exact mass, their elemental composition, and their fragmentation behavior (Appendix A), in comparison with standards and databases. The high-resolution accurate mass spectrometry analysis revealed the presence of 19 compounds, including phenolic acids and flavonoids in *A. borbonica* acetonic extracts (Table 2). Similar profiles were obtained from the *A. borbonica* aqueous extract (Appendix A).

Chromatographic peak 1 (0.52 min) (Figure 2A) showed a precursor ion [M − H]^−^ at *m*/*z* 191.0554 with the predicted molecular formula C_7_H_11_O_6_ (mass error 0.4 ppm), suggesting the presence of quinic acid. The MS2 spectrum (Appendix A) indicated a base peak at *m*/*z* 111.0076, associated with the successive loss of two water molecules and a -CO_2_ group from quinic acid.

Chromatographic peak 2 (2.17 min) showed a precursor ion [M − H]^−^ at *m*/*z* 153.0184, with the following composition of C_7_H_5_O_4_ (mass error 0.13 ppm) and a MS2 base peak at *m*/*z* 109.0283, resulting from the removal of a -CO_2_ group (Appendix A). Assignation to protocatechuic acid was achieved using a commercial standard.

Chromatographic peaks 3, 4, and 18 showed a precursor ion [M − H]^−^ at *m*/*z* 353.0878, with the following predicted molecular formula of C_16_H_17_O_9_ (mass error 1.03 ppm), suggesting the presence of caffeoylquinic acid isomers (CQA). Indeed, the MS2 spectra (Appendix A) show the same fragmentation pattern, with a base peak at *m*/*z* 191.0554 due to the loss of caffeic acid moiety, and the main product ions at *m*/*z* 179.0343, corresponding to loss of quinic acid moiety; *m*/*z* 173.0447, corresponding to water loss from quinic acid; and *m*/*z* 135.0441, corresponding to loss of a -CO_2_ group from caffeic acid. Among these three isomers, only peak 18 (6.2 min) had an MS2 base peak at *m*/*z* 173.0447, allowing the identification of 4-CQA, which is consistent with the 4-acylated mono-acyl CGAs [17] (Appendix A). Peaks 3 (2.63 min) and 4 (3.47 min) (Figure 2A) can be easily distinguished by their fragmentation. These peaks had both the same MS2 base peak at *m*/*z* 191.0554, which is consistent with 3-CQA and 5-CQA acylation but different intensities for the MS2 ion at *m*/*z* 179.0343, as previously reported [18,19,20]. They were identified as 3-CQA and 5-CQA, respectively (Appendix A).

Chromatographic peak 5 (3.68 min) had a precursor ion [M − H]^−^ at *m*/*z* 179.0350, with a predicted molecular formula C_9_H_7_O_4_ (mass error 0.2 ppm), suggesting the presence of caffeic acid. A MS2 base peak was observed at *m*/*z* 135.0441, corresponding to the loss of -CO_2_ group from caffeic acid. Furthermore, its identity was confirmed by comparing the fragmentation spectra and retention time of a caffeic acid reference standard.

Chromatographic peaks 6 (4.09 min) and 7 (4.18 min) showed a precursor ion [M − H]^−^ at *m*/*z* 337.0931, which has the predicted molecular formula C_16_H_17_O_8_ (mass error 0.1 ppm), corresponding to *p*-coumaroylquinic acid isomers (p-CoQA). These two peaks had the same MS2 base peak at *m*/*z* 191.0550 and secondary ions at *m*/*z* 173.0446 and 163.0392, corresponding to the dehydrated forms of quinic acid and coumaric acid, respectively. Peaks 6 and 7 were identified as 3- or 5-*p*-coumaroylquinic acids [21,22] (Appendix A).

Chromatographic peaks 8 (4.2 min) and 10 (4.43 min) with a precursor ion [M − H]^−^ at *m*/*z* 163.0391, which had the predicted molecular formula C_16_H_17_O_9_ (mass error 0.2 ppm), could be coumaric acid isomers. Indeed, these two peaks had the same MS2 ion at *m*/*z* 119.049, corresponding to the removal of a -CO_2_ group from coumaric acid. The identification was further confirmed by comparing the MS2 fragmentation behavior and the retention time of a *p*-coumaric acid reference standard. Therefore, peak 8 was identified as m/o-coumaric acid and peak 10 as *p*-coumaric acid (Appendix A).

For the chromatographic peak 9 (4.36 min), a precursor ion [M − H]^−^ at *m*/*z* 367.1035 with a predicted molecular formula C_17_H_19_O_9_ (mass error 0.5 ppm) was detected, suggesting the presence of feruloylquinic acid (FQA). The MS2 base peak at *m*/*z* 191.0550, associated with quinic acid and a product ion at *m*/*z* 173.0444, was often found for the 5-FQA [23] (Appendix A).

Chromatographic peak 11 (4.74 min) shows a precursor ion [M − H]^−^ at *m*/*z* 609.1464, with the following composition of C_27_H_29_O_16_ (mass error 1.6 ppm) and an MS2 base peak at *m*/*z* 300.0274, resulting from the neutral loss of a disaccharide rutinose linked to quercetin. Its identification as quercetin-3-*O*-rutinoside (rutin) was confirmed by comparing the MS2 fragmentation pattern and the retention time of its commercial standard (Appendix A).

Chromatographic peak 12 (4.94 min) and 13 (5.01 min) show a precursor ion [M − H]^−^ at *m*/*z* 463.0884, with the following composition of C_21_H_19_O_12_ (mass error 1.33 ppm) and an MS2 base peak at *m*/*z* 300.0274, resulting from the neutral loss of a hexose linked to quercetin. Their identification as quercetin-3-*O*-galactoside (hyperoside) (peak 12) and quercetin-3-*O*-glucoside (peak 13) (Appendix A) was solved by comparing their MS2 fragmentation pattern and their retention time to a hyperoside commercial standard that allowed a reliable discrimination.

For the chromatographic peaks 14 (5.26 min) and 15 (6.45 min), a precursor ion [M − H]^−^ at *m*/*z* 447.0935 with the predicted molecular formula C_21_H_19_O_11_ (mass error 1.35 ppm) was detected, suggesting the presence of kaempferol hexosides. The MS2 base peak at *m*/*z* 284.0326 was linked to the loss of the hexoside part, which reinforced their identification (Appendix A).

For the chromatographic peaks 16, 17, and 19, a precursor ion [M − H]^−^ at *m*/*z* 515.1195 with a predicted molecular formula of C_25_H_23_O_12_ (mass error 1.04 ppm) was detected, suggesting the presence of di-caffeoylquinic acid isomers (di-CQA). The main MS2 product ions were at *m*/*z* 353.0878, due to the loss of the caffeic acid moiety; *m*/*z* 191.0554, corresponding to quinic acid; *m*/*z*, 179.0342 corresponding to caffeic acid; *m*/*z* 173.0447, corresponding to a dehydrated quinic acid; and *m*/*z* 135.0440, corresponding to a decarboxylated form of caffeic acid. Interestingly, among these four isomers, only peak 16 (5.82 min) had an MS2 base peak at *m*/*z* 191.0554, allowing the identification of 3,5-diCQA (Appendix A). The other three isomers had an MS2 base peak at *m*/*z* 173.0447, which is consistent with the 4-acylated mono-acyl CGAs. In this way, the peak 17 (6.02 min) was assigned to 3,4-diCQA due to a higher intensity of the quinic acid product ion (*m*/*z* 191.0554) than the remaining peaks (Appendix A). Due to the lack of standards, peak 19 (6.36 min) was tentatively characterized as either 1,4-diCQA or 4,5-diCQA (Appendix A) [17,18]. Of note, for most of the identified compounds we found an 80% coverage with the MS2 spectra from the mzCloud Database.

#### Phenolic Acids Quantification by UHPLC-HESI-MS

The quantification by mass spectrometry highlighted a high abundance of cinnamic and benzoic acid derivatives in both extracts (Table 3, Appendix A). Most of the compounds were found in significantly higher concentrations in acetonic versus aqueous extracts: 0.002162 vs. 0.000703 mg/mL for caffeic acid, 0.007596 and 0.001437 mg/mL for dicaffeoylquinic acids isomers, and 0.004070 and 0.002415 mg/mL for protocatechuic acid. Interestingly, the concentration of caffeoylquinic acid isomers (5-CQA/3-CQA) was higher in the aqueous relative to the acetonic extracts (0.010163 vs. 0.005559 mg/mL). Indeed, the total amount of phenolic acids and flavonoids is higher in the acetonic extract than in the aqueous extract. These results highlight a notable amount of phenolic acids in both *A. borbonica* extracts.

Quantification by mass spectrometry confirmed the abundance of flavonol derivatives in both extracts (Table 3, Appendix A). Interestingly, we found three times more quercetin-3-*O*-rutinoside and quercetin-3-*O*-galactoside in the acetonic than in the aqueous extract, and five times more kaempferol-hexoxides in the acetonic than in the aqueous extract. These results confirmed the presence of notable amounts of flavonoids in both *A. borbonica* extracts.

### 2.3. Zebrafish Embryo and Larvae Acute Toxicity Test

#### Survival and Lethality Curves on Zebrafish Embryos

In order to investigate the toxic effect of polyphenols-rich acetonic and aqueous extracts from *A. borbonica*, a Fish Embryo Acute Toxicity (FET) test was performed according to the OECD guidelines [24]. Briefly, fertilized zebrafish eggs (0–3 hpf) were incubated with different concentrations of *A. borbonica* (acetonic or aqueous extract) until 96 hpf (developmental day 4), the extract being freshly renewed every day. Within the first day (0–24 hpf), none of the embryos survived at the highest concentrations of acetonic (16.9 g/L) and aqueous (40–22.5 g/L) extracts (Figure 3A,B) as shown by the coagulated eggs (Figure 4A, representative picture).

Obviously, the mortality rate is dose-dependent (Figure 3A–F). As shown in Figure 3C,D, the median lethal concentration (LC_50_) corresponding to the concentration that induced a 50% mortality was lower with the acetonic extract than with the aqueous extract (5 ± 0.2 vs. 17.6 ± 1.7 g/L, respectively), demonstrating the higher toxicity of the acetonic extract. At non-lethal concentrations (2.3 and 7.2 g/L for acetonic and aqueous extracts, respectively), incubation with polyphenols-rich acetonic and aqueous extracts from *A. borbonica* leads to developmental delay and malformations (Figure 4A–D). Although a 90% hatching was measured in the E3 medium, a significant decrease of 75% and 38% in hatching was observed at 96 hpf for the acetonic (2.3 g/L) and aqueous (7.2 g/L) extracts, respectively. This percentage reached 0% at 7.2 g/L (acetonic) and 9.5 g/L (aqueous) (Figure 4B). For the hatched embryos who have been exposed to the two types of extracts, we observed 21 ± 3% and 15 ± 1.6% pericardial edema with 2.3 g/L of acetonic and aqueous extracts, respectively, this percentage reaching 50% with 7.2 g/L of aqueous extract (Figure 4C). Spinal curvature was observed in 14 ± 2% and 15 ± 1.5% of these hatched embryos exposed, respectively, to acetonic and aqueous extracts, and reached 50% at 7.2 g/L of aqueous extract of *A. borbonica* (Figure 4A). Taken together, these data demonstrate the deleterious impact of such non-lethal concentrations during zebrafish development.

The toxicity of polyphenols-rich acetonic and aqueous extracts from *A. borbonica* was also studied in zebrafish larvae from 3 to 5 dpf. Indeed, at 3 dpf the swimming larvae already displayed functional livers and kidneys, allowing them to metabolize a variety of compounds [25,26,27]. As a consequence, the toxicity of the respective extracts could be different in zebrafish embryos and larvae. By treating the zebrafish larvae for 2 days in a way similar to the embryos, the respective LC_50_ values were determined for both extracts (Figure 3E,F). In embryos, the LC_50_ is higher with the aqueous extract than with the acetonic one (20.3 g/L vs. 5.6 ± 0.4 g/L). In addition, although no significant differences in LC_50_ were observed between the embryos and larvae, the LC_50_ is weakly higher for larvae than for embryos (20.3 g/L vs. 17.6 ± 1.7 g/L, respectively). Our data strongly suggest that even at non-lethal concentrations, these extracts can lead to developmental defects.

## 3. Discussion

Over the years, natural phenolic compounds have represented major preventive and/or therapeutic compounds for improving health issues. Indeed, many epidemiological studies have exhibited the beneficial effect of a polyphenol-rich diet on cancer, diabetes, obesity, and cardiovascular and neurodegenerative diseases [28,29,30,31,32,33,34,35]. Réunion island, a famous biodiversity hotspot, exhibits a wide and rich flora, with 22 medicinal plants registered to the French pharmacopeia [9]. These plants are known for their use in traditional medicine [3] and have been reported to be rich in polyphenols [36]. However, they are only poorly characterized concerning their contents, toxicities, and real in vivo preventive and/or therapeutic properties.

Among these medicinal plants, *A. borbonica*, belonging to the *Rubiaceae* family, is particularly interesting, as it widely used in traditional medicine for treating, among other thing, diabetes, urinary tract infections, diarrhea, hemorrhage, rheumatism, and also kidney stones [10]. Interestingly, in an ischemia-reperfusion stroke mouse model exposed to hyperglycemia, *A. borbonica* polyphenols display neuroprotective effects, preventing the elevation of the brain pro-inflammatory cytokine (IL-6) level and exerting its antioxidant property by decreasing reactive oxygen species (ROS) [13]. More recently, a preventive protective effect of *A. borbonica* aqueous extract was evidenced in a diet-induced overweight model in zebrafish displaying oxidative stress and blood–brain barrier leakage [14].

In this work, to the best of our knowledge, we performed for the first time an in-depth characterization of the polyphenol content of *A. borbonica* aqueous and acetonic extracts, demonstrating the presence of new molecules never described before for that plant. Furthermore, although used in humans, the safety profile of *A. borbonica* is largely unknown and no toxicological studies have been carried out so far. We consequently provided data concerning the toxicity of the *A. borbonica* aqueous and acetonic extracts on relevant in vivo physiological models using zebrafish embryos and larvae.

### 3.1. Polyphenol Content of Aqueous and Acetonic Extracts

A quantification by high-resolution mass spectrometry of the acetonic and aqueous extracts revealed the presence of polyphenol derivatives belonging to the phenolic acid and flavonoid classes known to be the most abundant in plant and plant-based foods [37,38]. From a qualitative point of view, no difference was observed between both extracts. Interestingly, although herbal infusion allows a very powerful polyphenol extraction yield, from a quantitative point of view the total amount of phenolic acids and flavonoids remains a little higher in the acetonic extract than in the aqueous extract. This result was expected, since it is well known that the polyphenol solubility depends on the solvent polarity and the kind of extraction used [39,40,41].

In this work, we have identified 19 main polyphenols. Among them, we observed that the major compounds of both extracts were quercetin-3-*O*-rutinoside, caffeoyl- and dicaffeoyl-quinic acids isomers, protocatechuic acid, coumaric acids isomers, and caffeic acid. These results were consistent with previous ones reported from our laboratory [5]. However, we provide a significant input to this previous work, since our MS^2^ spectral analysis identified new compounds such as quercetin-3-*O*-rutinoside, protocatechuic acid, and coumaric acids isomers.

The identification of these new compounds could be explained by a different geographical location of *A. borbonica* leading to different environmental conditions, such as moisture, illumination, altitude, and temperature [42]. Furthermore, in these previous studies, conventional C18 reverse-phase column was used, while in the present study we used a pentafluorophenyl-phase column known to offer a greater selectivity for several compound classes such as aromatic and isomeric compounds, achieved not only by hydrophobic interactions as C18 reverse-phase but also by aromatic, π–π, dipole–dipole, and ionic interactions and hydrogen bonding [43].

High phenolic acid and flavonoid contents are often associated with a high antioxidant capacity of the plant extract [44,45]. These compounds are characterized by one or several aromatic rings with at least one hydroxyl group [37]. This particular chemical structure conferred them different antioxidant activities due to their ability to directly scavenge free radicals by donating protons/electrons [46,47] or by activating antioxidant signaling pathways [48]. Interestingly, the DPPH experiment highlighted the strong capacity of both extracts to reduce DPPH radicals due to their high content in bioactive molecules. These high antioxidant properties for the acetonic extract are consistent with a previous study [5].

### 3.2. Aqueous and Acetonic Extracts of Antirhea borbonica Exhibit Developmental and Toxicity at High Concentrations

Despite the beneficial effects of natural polyphenols, the safe consumption of beverages from medicinal plants remains poorly studied. Indeed, polyphenols can also display adverse effects, including carcinogenic/genotoxic ones; act as endocrine disruptors; disturb iron absorption; and also interact with drugs [49]. We consequently decided to investigate for the first time the potential toxicity of the medicinal plant *A. borbonica* using the zebrafish model. Whilst the classical approach for the assessment of drug and plant extract toxicity is time consuming, expensive, and requires in vitro and in vivo models (mainly rodents), the zebrafish embryo model emerged as a relevant tool for a first toxicological screening approach. This developmental model is widely used to assess drug toxicity [15] and appears as a relevant model, given its strong genic homology (more than 70% of human genes have an orthologue in zebrafish). As well, zebrafish display many evolutionary conserved organs and physiological processes [16] and exhibit a high fertility rate as well as transparent eggs, allowing their easily monitoring at the different stages of organogenesis [50].

In the present study, we used the Fish Embryo Acute Toxicity test (FET) designed by OECD and known to be a reference in the field [24]. At the highest doses tested (40–16.9 g/L), a 100% mortality was observed at 24 hpf for both extracts, in spite of the chorion presence, which potentially acts as a barrier. The toxicity of both plant extracts evidenced that the acetonic extract was more toxic than the aqueous extract (LC_50_: 5 ± 0.2 g/L vs. 17.6 ± 1.7 g/L, respectively). Such a toxicity may be associated with the high bioactivity of the plant extracts or the presence of other phytochemical compounds, such as alkaloids and terpenes, known to be produced for defense against abiotic and probiotic stresses in plants. Previous studies have reported that alkaloids significantly contribute to the toxic effect of various plants [51,52,53].

Interestingly, for the highest non-lethal concentrations (2.3 g/L (acetonic), 7.2 g/L (aqueous)), we observed a delay in hatching in a concentration- and exposure time-dependent manner. A significant hatching reduction of 38% and 75% was observed at 96 hpf for the aqueous (7.2 g/L) and acetonic (2.3 g/L) extracts, respectively. Hatching normally occurs between 72 and 96 hpf. Although this remains to be investigated, we can speculate that the delayed hatching observed in our experimental conditions for the treated embryos may be induced by an impairment in the hatching process, such as the production-secretion of hatching enzymes (zhe1), cathepsin L (catL1), involved in this physiological process [54], as well as in the expression of genes (Zip10 and Znt1a) involved in zinc metabolism that are known to be essential in the development of the hatching gland [55]. In the chorion, a decrease in larvae motricity due to a delay in the muscular development or an altered larval morphology may also explain the delayed hatching.

As well, we noticed several malformations such as spinal curvature and the presence of pericardium edema. These defects are known to occur in the presence of toxic molecules, leading to kidney impairments or kidney malformations [56,57,58]. Further investigations are needed to identify the presence of compounds or metabolites inside the zebrafish embryos responsible for the acute toxicity of *A. borbonica* extracts.

We also aimed at identifying the potential toxicity of our extracts in larvae (from 3 to 5 dpf). At 3 dpf, zebrafish larvae exhibit functional livers and kidneys, suggesting the possible metabolism of several compounds [59,60]. Interestingly, the study of the toxicity of polyphenols-rich acetonic and aqueous extracts from *A. borbonica* in zebrafish larvae from 3 to 5 dpf revealed no significant differences in the median lethal concentrations between the larvae and embryos, with 5 ± 0.2 vs. 5.6 ± 0.4 g/L and 17.6 ± 1.7 vs. 20.3 g/L for acetonic and aqueous extracts, respectively.

Medicinal plants known as “green gold” have been widely used in traditional medicine for centuries across the world. The relatively toxic effect of the medicinal plant relies on different parameters, such as the plant part used, composition, preparation method, and concentration. Among plant-based beverage preparations, the infusion is the most commonly used after decoction. Therefore, we studied the potential toxic effect of an aqueous extract from *A. borbonica* on zebrafish embryos. A median lethal concentration of 17.6 ± 1.7 g/L was found. This concentration is higher than the concentration recommended by the herbalist for this method of preparation, around 1–4 g for 1 L. However, we must remain vigilant about taking this aqueous extract, which can have teratogenic effects, especially during pregnancy.

## 4. Materials and Methods

### 4.1. Reagents/Standards

Folin-Ciocalteu reagent, sodium carbonate, sodium nitrite, aluminum chloride, DPPH, caffeic acid, caffeoyquinic acid, *p*-coumaric acid, protocatechuic acid, kaempferol, hyperoside, rutin hydrate, and dimethylsufoxide (DMSO) were purchased from Sigma Aldrich (St. Louis, MO, USA). Solvents such as acetone, acetonitrile, methanol. and water (HPLC-MS grade) were purchased from Carlo erba (Peypin, France).

### 4.2. Plant Material

*Antirhea borbonica* J.F Gmelin (*A. borbonica*) powder (leaves) was obtained from APLAMEDOM institute (Association pour les plantes aromatiques et médicinales de La Réunion) and registered under the following code, DéTROI.002/2018, stating the date of collection and the GPS coordinates (21°05′44.9″ S, 55°39′06.6″ E), altitude: 770 m. The pharmacist and director of APLAMEDOM performed the botanical identification of *A. borbonica*. The dried (air-dried protected from direct light) leaves were reduced to powder using a laboratory grinder. The crushed leaves of *A. borbonica* were conserved at −20 °C.

### 4.3. Preparation of the Plant Extracts

Acetonic extract from *A. borbonica* was obtained after dissolving 1 g of crushed leaves in 25 mL of an aqueous acetonic solution (70%, *v/v*). After incubation at 4 °C for 90 min, the mixture was centrifuged at 3500 rpm at 4 °C for 20 min and the polyphenol-rich supernatant was collected and dried using a rotary evaporator. The extract was resuspended with an identical volume of E3 medium (classical embryonic medium), filtered with a 20 µm membrane, and stored at −20 °C until analysis.

Aqueous plant extract (40 g/L) was prepared by the infusion technique. Briefly, 1 g of crushed leaves was added to 25 mL of boiled E3 medium for 10 min under stirring. The resultant extract was filtered with a 20 µm membrane, aliquoted, and stored at −20 °C until analysis.

For a toxicity test using fertilized eggs and larvae zebrafish, aqueous extract was prepared every day extemporaneously.

### 4.4. Measurement of the Total Antioxidant Capacity of Polyphenol-Rich Plant Extracts

The total antioxidant capacity was assessed using the 2,2-Diphenyl-1-picrylhydrazyl (DPPH) radical scavenging assay. Briefly, in a 96-well microplate, 200 µL of a 0.25 mM DPPH solution and 40 µL from different concentrations (40–2.5 g/L (GAE)) of acetone-evaporated or aqueous extracts were added and incubated at 25 °C for 30 min. Ascorbic acid solutions prepared at the same concentration range (gallic acid equivalent) were used as an antioxidant standard. The absorbance (Abs) was read at 517 nm (FLUOstar Optima, Bmg Labtech, Offenburg, Germany). The percentage of the free radical-quenching activity of DPPH was calculated from the following formula:(1)$$\mathrm{Antioxidant}\;\mathrm{capacity}\;(\%)=[({\mathrm{Abs}}_{\mathrm{control}}-{\mathrm{Abs}}_{\mathrm{sample}})/{\mathrm{Abs}}_{\mathrm{control}}]\times 100.$$


Inhibitory concentration (IC_50_) values corresponding to the concentrations reducing 50% of the initial DPPH• values were obtained by plotting the percentage of free radical-quenching activity against the logarithm of the different concentrations ranging from 16.9 to 1.4 g/L (acetonic) and 40 to 2.5 g/L (aqueous). Once the concentration values were transformed, a nonlinear regression (log (inhibitor) vs. response-variable slope) was applied to obtain a sigmoid curve.

### 4.5. Determination of Phenolic Acid Content

The total phenolic acid contents in acetone-evaporated and aqueous extracts were determined by using the Folin–Ciocalteu assay [61] with slight modifications. Briefly, in a 96-well microplate, 25 μL of plant extract, 125 μL of Folin-Ciocalteu’s phenol reagent (Sigma Aldrich), and 100 μL of 75 g/L sodium carbonate (Sigma) were added and incubated at 50 °C for 15 min and then at 4 °C for 3 min. The absorbance was measured at 760 nm (FLUOstar Optima, Bmg Labtech). A calibration curve between 12.5 and 300 µM was prepared using a standard solution of gallic acid (Sigma-Aldrich, Darmstadt, Germany). The total phenolic acid contents were expressed as the mg gallic acid equivalent (GAE) per 100 g of dried plant powder.

### 4.6. Determination of Flavonoid Content

The total flavonoid content was determined using the aluminum chloride (AlCl_3_) colorimetric assay adapted from Zhishen et al. [62]. For this measurement, 100 μL of sample was mixed in a 96-well microplate with 6 μL of 5% aqueous sodium nitrite (NaNO_2_) solution. After 5 min, 6 μL of 10% aqueous AlCl_3_ were added and the mixture was vortexed. Then, after 1 min incubation, 40 μL of 1 M NaOH was added. The absorbance was read at 510 nm (FLUOstar Optima, BMG Labtech). A calibration curve between 6.25 and 300 µM was prepared using a standard solution of quercetin. The total flavonoid contents were expressed as the mg quercetin equivalent (QE) per 100 g of dried plant powder.

### 4.7. Polyphenolic Compounds Identification and Quantification LC-UV-HESI-MS/MS

Polyphenols extracted from *A. borbonica* acetone-evaporated or aqueous extracts were identified by ultra-high-performance liquid chromatography, coupled with diode array detection and HESI-Orbitrap mass spectrometer (Q-Exactive™ Plus, Thermo Scientific, Waltham, MA, USA). Briefly, 10 µL of the sample was injected using an UHPLC system equipped with a Thermo Fisher Ultimate 3000 series WPS-3000 RS autosampler and then separated on a PFP column (2.6 μm, 100 mm × 2.1 mm, Phenomenex, Torrance, CA, USA). The column was eluted with a gradient mixture of 0.1% formic acid in water (A) and 0.1% formic acid in acetonitrile (B) at the flow rate of 0.450 mL/min, with 5% B at 0.00 to 0.1 min, 35% B at 0.1 to 7.1 min, 95% B at 7.2 to 7.9 min, and 5% B at 8.0 to 10 min. The column temperature was held at 30 °C and the detection wavelengths were set to 280 and 310 nm, allowing the identification of phenolic acids and flavonoids, respectively.

For the mass spectrometer conditions, a Heated Electrospray Ionization source II (HESI II) was used. Nitrogen was used as the drying gas. The mass spectrometric conditions were optimized as follows: spray voltage 2.8 kV, capillary temperature 350 °C, sheath gas flow rate 60 units, aux gas flow rate 20 units, and S lens RF level 50. Mass spectra were registered in full scan mode from *m*/*z* 100 to 1500 in negative ion mode at a resolving power of 70,000 FWHM at *m*/*z* 400. The automatic gain control (AGC) was set at 1e6. The MS/MS spectra were obtained by applying a relative higher energy collisional dissociation (HCD) energy of 25%. The identification of the compounds of interest was based on their retention time, exact mass, elemental composition, MS fragmentation pattern, and comparisons with available standards and the advanced mass spectral database, *m*/*z* Cloud, https://www.mzcloud.org. Data were acquired with the XCalibur 4.1 software (Thermo Fisher Scientific Inc.) and processed with the compound discoverer 2.1 and the Skyline 20.1 software (MacCoss Lab.) 1 × 10^6^.

#### Preparation of Standard Solution, Calibration Curves, and Method Validation

Standard stock solutions of caffeic acid, caffeoylquinic acid, kaempferol, quercetin-3-*O*-rutinoside, quercetin-3-*O*-galactoside, protocatechuic acid, and coumaric acid were dissolved in methanol at a concentration of 1 mg/mL. A mixed stock solution containing 10 µg/mL of each polyphenol standard was prepared in methanol. The calibration standard solutions were prepared by the dilution of the mixed stock solutions in 0.1% formic acid in water to obtain the desired calibration curves ranging from 10 to 4000 ng/mL. The quality control (QC) samples were prepared at 25, 250, and 4000 ng/mL and analyzed in triplicate within each batch.

The calibration curves were built by plotting the peak area of the analytes against the corresponding analytes concentrations with linear regression using standard samples at nine concentrations. The calibration curves of each polyphenol had a correlation coefficient (R2) of 0.99. The method accuracy was estimated by calculating the percent deviation observed in the analysis of QC samples and expressed by relative error. The intraday precision was estimated by analyzing QC samples at three concentration levels (25, 250, 4000 ng/mL) of the seven analytes within 24 h (n = 8). The inter-day accuracy was estimated by the repeated analysis of QC samples (n = 8).

The variability was expressed as the relative standard deviation (RSD, %), and the accuracy was expressed as the relative error (RE, %). The limit of quantification (LOQ) was defined as the lowest analytical concentration of the calibration curve at which the measured precision, expressed as relative standard deviation (RSD), was within 20% and the accuracy, expressed as relative error (RE), was in the range of 20%.

### 4.8. Zebrafish Husbandry

Adult AB wildtype zebrafish (Danio rerio, AB strain) were housed in the zebrafish facility of the CYROI/DéTROI, La Réunion (A974001). They were maintained under the standard conditions of photoperiod (14 h dark/10 h light), temperature (28.5 °C), conductivity (400 μS), and pH (7.4). Zebrafish were fed daily (3 times a day) with commercially available food (Planktovie, GEMMA 300). All the animal experiments were performed in CYROI/DéTROI (UMR 1188) and conducted in accordance with the French and European Community Guidelines for the Use of Animals in Research (86/609/EEC and 2010/63/EU).

### 4.9. Developmental Toxicity Test (Zebrafish Embryos)

The day before the start of the toxicity test assay, breeding males and females (optimal ratio 2:1) were placed in the same tank but were physically separated. The next day, 1 h after light onset, fish couples were allowed to spawn for 1 h. The eggs were collected, rinsed with fish water system, randomly mixed, and quickly distributed in the different concentrations of *A. borbonica* aqueous and acetone-evaporated extracts prepared with E3 medium. The treatment was performed between 1 and 3 h post fertilization (dpf) for embryos and between 3 and 5 dpf for larvae. The quality of the spawn (>70% of fertilization) was checked for fitting with the OECD recommendations (guidelines 236: Fish Embryo Acute Toxicity (FET) Test) [24]. The fertilized eggs were selected using a stereomicroscope and dispatched in a 24-well plate as follows: in each well, five fertilized eggs were placed in 2 mL of the respective concentrations of either *A. borbonica* aqueous extract (40, 30, 22.5, 16.9, 12.7, 9.5, 7.2, and 2.3 g/L) or *A. borbonica* acetone-evaporated extract (16.9, 12.7, 9.5,7.2, 5.4, 4, 3, 2.3, 1.7, and 1.3 g/L) diluted in E3 medium. The concentration range was chosen on the basis of traditional use, which consists of infusion of 1–4 g of dried leaves in 1 L of boiled water for 10 min. Because *A. borbonica* has been registered at the French pharmacopeia, this concentration was supposed to be non-toxic in adults. Thus, the lowest doses tested were 2.3 and 1.3 g/L for aqueous and acetone-evaporated extract, respectively.

A total of 20 eggs or 10 larvae were tested for each concentration. These experiments were repeated three times independently. The 24-well plate was incubated at 26 °C ± 1 °C. Negative controls (E3 medium only) and positive controls (E3 medium + 25% DMSO) were also placed in the 24-well plate.

The treatment was renewed each day using a freshly prepared *A. borbonica* extracts. Zebrafish development was carefully checked by using a stereo microscope (Nikon SMZ18) at 24, 48, 72, and 96 hpf looking at four apical observations as indicators of lethality according to the OECD guidelines 236: (i) coagulation of fertilized eggs, (ii) lack of somite formation, (iii) lack of detachment of the tail-bud from the yolk sac, and (iv) lack of heartbeat. At the end of the exposure period, the acute and developmental toxicity (teratogenicity) were determined according to the OECD ruled based on a positive outcome in any of the four apical observations recorded. The percentage of mortality was determined by using the following equation:(2)(Mortality (%) = (Number of dead embryos/Total number of embryos)×100.


Lethal concentrations (LC_50_) corresponding to the concentration that induced a 50% mortality were obtained by plotting the percentage of cumulative mortality at 96 hpf against logarithm of the different concentrations. A nonlinear regression (log (inhibitor) vs. response-variable slope was applied to obtain a sigmoid curve. In addition, morphological abnormalities such as spinal curvature, delay in pigmentation, delay in eye color, and delay hatching were recorded.

### 4.10. Statistical Analyses

Data are expressed as the mean ± standard deviation (SD) from at least three independent experiments performed in triplicate. Statistical analyses and determination of IC_50_/LC_50_ were performed with Graph-Pad Prism 6.3 (GraphPad Software, Inc., San Diego, CA, USA). Comparison between more than 2 groups was determined using a one-way ANOVA followed by Dunnett’s test. A *p*-value < 0.05 was considered statistically significant.

## 5. Conclusions

In this study, we identified new major polyphenols such as quercetine-3-*O*-rutinoside, protocatechuic acid, and coumaric acids isomers, which have to be taken into account regarding the anti-oxidant and anti-inflammatory effect of *A. borbonica*. We report for the first time the potential embryonic and larval toxicity at high concentrations of both acetonic and aqueous extracts from the medicinal plant *A. borbonica* by using the zebrafish embryo model. The present work will be useful to supplement current data on medicinal plants registered at the French pharmacopeia and more generally can be considered as a “proof of concept study” for the further analysis of medicinal plants.

## Figures and Tables

**Figure 1 molecules-25-04482-f001:**
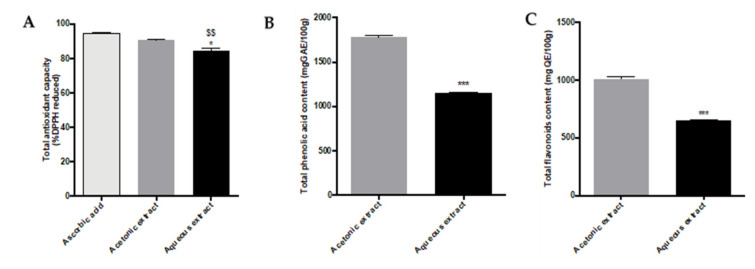
Antioxidant capacity, total phenolic acid, and flavonoid contents from *A. borbonica* extracts. (**A**) Total antioxidant capacity of polyphenols-rich extracts from *A. borbonica* was measured by a the 2,2-Diphenyl-1-picrylhydrazyl (DPPH) assay. Ascorbic acid was used as a positive control. The results are expressed as the % of reduced DPPH. (**B**) Total phenolic contents of acetonic and aqueous extracts from *A. borbonica* were determined using the Folin–Ciocalteu colorimetric assay at a concentration of 40 g/L (plant dried powder). The results are expressed as the mg gallic acid equivalent (GAE)/100 g of plant dried powder. (**C**) Total flavonoid contents were determined using an aluminum chloride colorimetric assay. The results are expressed as the mg quercetin equivalent (QE)/100 g of plant dried powder. Data are the means ± SDs of three independent experiments. * *p* < 0.05, *** *p* < 0.001 (vs. 40 g/L of acetonic extract), and ^$$^
*p* < 0.01 (vs. 40 g/L (GAE) ascorbic acid).

**Figure 2 molecules-25-04482-f002:**
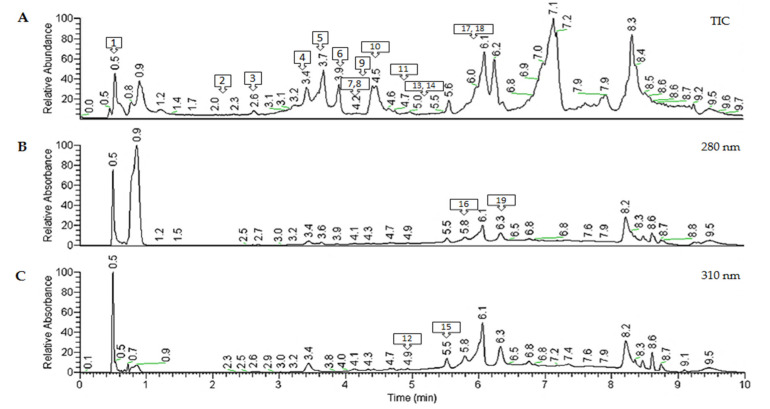
Spectra obtained for a representative *A. borbonica* acetone-evaporated extract. (**A**) Representative total ion chromatogram (TIC) obtained in negative mode. (**B**) UHPLC-UV chromatograms obtained at 280 and 310 nm (**C**). The different molecules are numbered according to their retention times.

**Figure 3 molecules-25-04482-f003:**
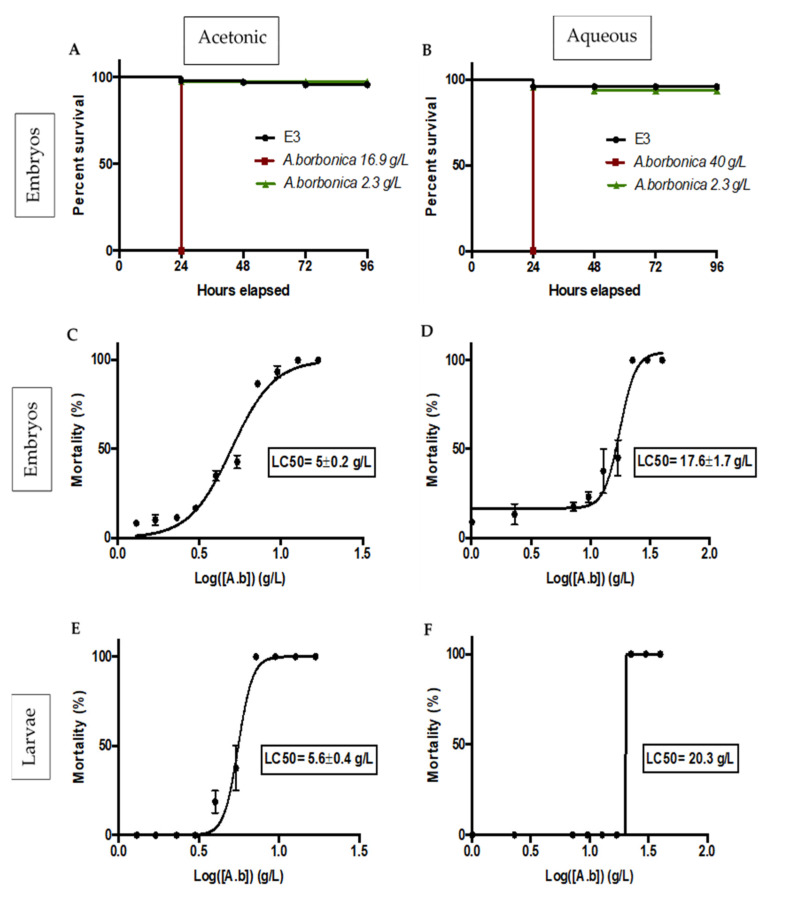
Survival curves for 96 hpf zebrafish embryos exposed with acetonic (**A**) or aqueous (**B**) extracts from *A. borbonica* at high concentrations of 16.9 g/L (acetonic) and 40 g/L (aqueous) and a low concentration of 2.3 g/L (acetonic and aqueous), and E3 was considered as control. Median lethal concentration curves (LC_50_) for 96 hpf zebrafish embryos and 72 hpf larvae exposed to acetonic (**C**,**E**) or aqueous (**D**,**F**) extracts at different concentrations ranging from 16.9 to 1.3 g/L (acetonic) and 40 to 2.3 g/L (aqueous) for 4 and 2 days, respectively. The LC_50_ values were expressed in g/L. Data are the mean ± SD of three independent experiments.

**Figure 4 molecules-25-04482-f004:**
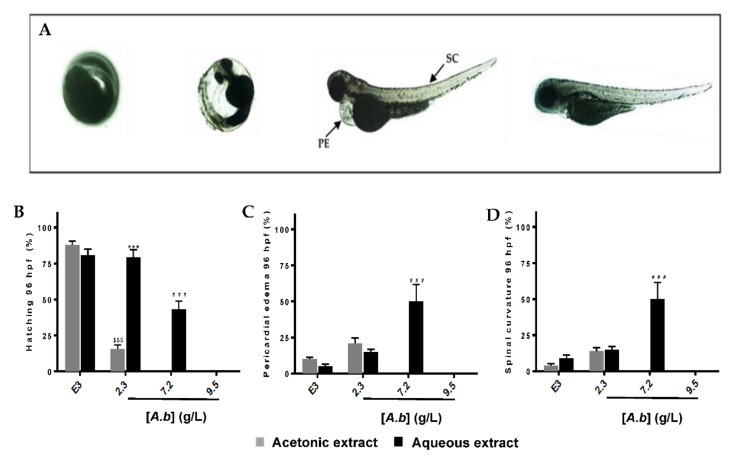
Morphological malformations and delayed development of zebrafish embryos/larvae exposed to *A. borbonica* extracts. (**A**) From left to right, coagulated egg (at 24 hpf), delayed hatching, spinal curvature, pericardial edema, and control embryos/larvae (96 hpf). Arrows indicate the presence of pericardial edema (PE) and spinal curvature (SC). Hatchability rates after 4 days of exposure with acetonic and aqueous extracts at 2.3, 7.2, and 9.5 g/L are represented in (**B**–**D**) represent the percentage of PE and SC, respectively. E3 medium was used as a positive control. Data are the mean ± SD of three independent experiments. ^$$$^
*p* < 0.001 (vs. E3 (acetonic)), *** *p* < 0.001 (vs. acetonic extract), and ^###^
*p* < 0.001 (vs. E3 (aqueous)).

**Table 1 molecules-25-04482-t001:** Antioxidant activities of polyphenols-rich extracts from *A. borbonica* were measured by DPPH assay. Ascorbic acid was used as a positive control. The IC_50_ values were obtained by plotting the percentage of free radical-quenching activity against the logarithm of the different concentrations, ranging from 40 to 2.3 g/L (plant dried powder) for aqueous and acetonic extracts. The results were expressed in g/L. Data are the mean ± SD of three independent experiments.

IC_50_ (g/L)		
Ascorbic Acid	Acetonic Extract	Aqueous Extract
2.8 ± 0.1	3.1 ± 0.3	3.3 ± 0.3

**Table 2 molecules-25-04482-t002:** Identification of 19 compounds in the *Antirhea borbonica* acetonic extract by LC-UV-HESI-MS/MS in negative mode.

Peak Number	RT (min)	Compound	Molecular Formula	Mass Error (ppm)	[M − H]^−^	MS/MS Fragments	mzCloud Best Match (%)
1	0.52	d-Quinic acid	C_7_H_12_O_6_	0.4	191.0554	111.0076	85.5
2	2.17	Protocatechuic acid	C_7_H_6_O_4_	0.13	153.0184	109.0283	82.7
3	2.63	3-Caffeoylquinic acid	C_16_H_18_O_9_	1.03	353.0877	191.0554, 179.0343, 173.0447, 135.0441	85
4	3.47	5-Caffeoylquinic acid	C_16_H_18_O_9_	1.03	353.0877	191.0554, 179.0343, 173.0447, 135.0441	88.3
5	3.68	Caffeic acid	C_9_ H_8_ O_4_	0.2	179.0341	135.0441	80.2
6	4.09	*p*-Coumaroyl quinic acid isomer	C_16_H_18_O_8_	1.3	337.0931	191.0550, 173.0446, 163.0392	84.6
7	4.18	*p*-Coumaroyl quinic acid isomer	C_16_H_18_O_8_	1.3	337.0931	191.0550, 173.0446, 163.0392	84.6
8	4.2	*o/m*-Coumaric acid	C_9_H_8_O_3_	0.2	163.0391	119.049	81.2
9	4.36	Feruloylquinic acid	C_17_H_20_O_9_	0.5	367.1035	191.0550, 173.0446	_
10	4.43	*p*-Coumaric acid	C_9_H_8_O_3_	0.1	163.0391	119.049	81.2
11	4.74	Quercetin-3-*O*-rutinoside (Rutin)	C_27_H_30_O_16_	1.6	609.1466	300.0274	94.8
12	4.94	Quercetin-3-*O*-galactoside	C_21_H_20_O_12_	1.33	463.0884	300.0274	90.9
13	5.01	Quercetin-3-*O*-glucoside	C_21_H_20_O_12_	1.33	463.0884	300.0274	90.9
14	5.26	Kaempferol-*O*-hexoside	C_21_H_20_O_11_	1.35	447.0935	284.0326	83.7
15	5.45	Kaempferol-*O*-hexoside	C_21_H_20_O_11_	1.35	447.0935	284.0326	83.7
16	5.82	3,5-Dicaffeoylquinic acid	C_25_H_24_O_12_	1.04	515.1196	353.0878, 191.0554, 179.0343, 173.0447, 135.0441	83.6
17	6.02	3,4-Dicaffeoylquinic acid	C_25_H_24_O_12_	1.04	515.1195	353.0878, 173.0447, 191.0554, 179.0343, 135.0441	88.1
18	6.2	4-Caffeoylquinic acid	C_16_H_18_O_9_	1.03	353.0877	173.0447, 191.0554, 179.0343, 173.0447, 135.0441	86.3
19	6.36	1,4/4,5-Dicaffeoylquinic acid	C_25_H_24_O_12_	1.04	515.1194	353.0878, 173.0447, 191.0554, 179.0343, 135.0441	89.1

**Table 3 molecules-25-04482-t003:** Quantification of polyphenols-rich acetonic and aqueous extracts from *A. borbonica* by HPLC-HESI-MS. The analysis was performed using a Q-Exactive™ Plus mass spectrometer at a concentration of 40 g/L. The concentrations of the different compounds were expressed as ng/mL. Data are the mean ± SD of three independent experiments. * *p* < 0.05, ** *p* < 0.01 ***, *p* < 0.001 (vs. 40 g/L of acetonic extract). CQA: caffeoylquinic acid. Di-CQA: dicaffeoylquinic acid.

		Concentration in Acetonic Extract (mg/mL)	Concentration in Aqueous Extract (mg/mL)
Peak		Phenolic Acids	
5	Caffeic acid	0.002162 ± 0.000066	0.000703 ± 0.000039 ***
10	*p*-Coumaric acid	0.002755 ± 0.000728	0.001768 ± 0.000176 *
8	*m/o*-Coumaric acid	0.000470 ± 0.000003	0.000208 ± 0.000004
4	5-CQA	0.004718 ± 0.000279	0.008558 ± 0.000477 ***
3	3-CQA	0.000840 ± 0.000093	0.001604 ± 0.000157 ***
17	3,4-diCQA	0.004704 ± 0.000326	0.000503 ± 0.000034 ***
19	1,4/4,5-diCQA	0.000262 ± 0.000020	0.000090 ± 0.000003 **
16	3,5-diCQA	0.002629 ± 0.000161	0.000842 ± 0.000029 ***
2	Protocatechuic acid	0.004070 ± 0.000250	0.002415 ± 0.000387 ***
	Total	0.023061	0.016693
		**Flavonols**	
11	Quercetin-3-*O*-rutinoside	0.011933 ± 0.002018	0.003977 ± 0.000473 ***
12	Quercetin-3-*O*-galactoside	0.001791 ± 0.000204	0.000591 ± 0.000033 ***
14/15	Kaempferol hexosides	0.000216 ± 0.000054	0.000044 ± 0.000005 **
	Total	0.013941	0.004612

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
