# Peer review of "Phenolic Profile of Herbal Infusion and Polyphenol-Rich Extract from Leaves of the Medicinal Plant Antirhea borbonica: Toxicity Assay Determination in Zebrafish Embryos and Larvae"

_molecules, 2020, doi:10.3390/molecules25194482_

Round 1
Reviewer 1 Report
The authors present a study dedicated to the chemical characterization of Antirhea borbonica by using different chemical techniques and by using an in vivo toxicological model, the zebrafish embryo test. Overall, this is an interesting, new, and original work presenting novel chemical research on this topic which was further validating using an in vivo toxicological model.
However, some issues should be addressed:
- Review the use of italics for species and latin terms.
- Include statistics and p-values in the text.
- Change the order of the results section to meet the same order presented in the methods.
- Review the molecular formulas shown in Table 2 and in text. Numbers should be below the line. The same for LC50.
- Assign peak numbers to the compounds shown in Table 3. Also, change que quantification to mg/mL and include a total for the quantification in the last line of the table.
- Remove references from the results section.
- Include a figure with the results from the malformations observed rather than including an exemplificative figure (Fig 4A).
- Figure 4B is missing.
- The references 35 and 36 should be included in the introduction of the manuscript.
- The references used to describe zebrafish (50-52) should be changed by those included in the introduction.
- Include the concentration of sodium carbonate used for the total phenolic acid contents.
- Review the reference style.
Author Response
Reviewer 1
We thank the reviewer for valuable suggestions.
1 Review the use of italics for species and Latin terms.
We have reviewed the Latin terms and put in italics “A borbonica”, “in vivo”, “in vitro”, “Rubiaceae”. I would like to clarify that in the version we had submitted the Latin terms were in italics.
2 Include statistics and p-values in the text.
We have added p-values when appropriated
3 Change the order of the results section to meet the same order presented in the methods.
In fact, we thought it was more rational to change the order in the methods section. Paragraph 4.4 is now 4.5; paragraph 4.5 is now 4.6 and paragraph 4.6 is now 4.4
4 Review the molecular formulas shown in Table 2 and in text. Numbers should be below the line. The same for LC50.
In table 2 molecular numbers are now below the line as well LC50. Similar modification has been made in table S1.
5 Assign peak numbers to the compounds shown in Table 3. Also, change que quantification to mg/mL and include a total for the quantification in the last line of the table.
Peak numbers have been assigned to compounds in Table 3 and data are expressed now in mg/ml.
6 Remove references from the results section.
In the result section 2.2 it is not possible to remove the references because we have used these references to identified some compounds and those references are not cited elsewhere.
7 Include a figure with the results from the malformations observed rather than including an exemplificative figure (Fig 4A).
We agree with the reviewer that it would be easier to understand this part with additional graphs. So, we have added Figure 4C (pericardial edema) and 4D (spinal curvature). However, because we do believe that representative images allow a better understanding about what those malformations looks like, we let figure 4 A.
We thus rewrote this part: “At non-lethal concentrations (2.3 g/L and 7.2 g/L for acetonic and aqueous extracts, respectively), incubation with polyphenols-rich acetonic and aqueous extracts from A. borbonica leads to developmental delay and malformations (Figure 4A-D). Although 90% hatching was measured in the E3 medium, a significant decrease of 75% and 38% in hatching was observed at 96hpf for the acetonic (2.3 g/L) and aqueous (7.2 g/L) extracts, respectively. This percentage reached 0% at 7.2 g/L (acetonic) and 9.5 g/L (aqueous) (Figure 4B). For the hatched embryos, who have been exposed to the two types of extracts we observed 21 ± 3 % and 15 ± 1.6% of pericardial edema with 2.3g/l of acetonic and aqueous extracts respectively, this percentage reaching 50% with 7.2gL of aqueous extract (Figure 4C). Spinal curvature was observed in 14 ± 2 % and 15 ± 1.5 % of these hatched embryos exposed respectively to acetonic and aqueous extracts and reached 50% at 7.2 g/l of aqueous extract of A. borbonica (Figure 4D). Taken together these data demonstrate the deleterious impact of such non-lethal concentrations during zebrafish development.”
8 Figure 4B is missing.
We put the figure 4B back in the manuscript. We’re very sorry for that problem but again figure 4B was in the version we submitted. Probably a problem with the website!
9 The references 35 and 36 should be included in the introduction of the manuscript.
Both references have been now included in the introduction section of the manuscript are now reference 13 and 14: “as well as, in vivo, in a mouse stroke model [publi 35 is now 13] and a diet-induced overweight zebrafish model [publi 36 is now 14]. Importantly these antioxidant and anti-inflammatory biological effects were associated to the capacity of polyphenols to regulate key molecular targets such as ROS-producing and detoxifying enzymes, the redox-sensitive translational factor Nrf2 and improve vasoactive markers in these in vitro and in vivo pathological models [5,11,12, 13, 14].”
10 The references used to describe zebrafish (50-52) should be changed by those included in the introduction.
References 50, 51and 52 have been deleted and replaced by references 15 and 16 in the discussion part.
11 Include the concentration of sodium carbonate used for the total phenolic acid contents.
The concentration of sodium carbonate (75 g/l) has been added.
12 Review the reference style.
We have reviewed reference style according to molecules rules. Here again the reference style was that of molecules in the manuscript submitted.
Reviewer 2 Report
My comments and suggestions are highlighted within the file, as well as some relevant comments were placed within text boxes. I hope this review serves the authors.
Best Regards

Author Response
We thank the reviewer for the suggestions that have been taken into account in the revised version as shown by the tracking mode in the revised version. docs document.
Reviewer 3 Report
The authors studied thoroughly the aqueous and acetonic extracts of leaves of Antirhea borbonica. Using selective liquid chromatography-tandem mass spectroscopy they identified 19 polyphenols. They determined the toxicity of the extracts (LC50 values) in vivo by an assay in zebrafish embryos and larvae, which is valuable contribution to the knowledge on the safety profile of A. borbonica. The detailed description of the experiments, photos and large number of references show, that the authors are experts in this field.
The latin names e.g. Antirhea borbonica, A. borbonica, or in vivo, etc. should be written in Italic letters. Some abbreviations (e.g. DPPH line 66.) are not resolved in the manuscript.
Line 491.: Fish Embryo Acute Toxicity (FET) Test is the reference 22 [22].
Author Response
The latin names e.g. Antirhea borbonica, A. borbonica, or in vivo, etc. should be written in Italic letters. Some abbreviations (e.g. DPPH line 66.) are not resolved in the manuscript.
All the corrections have been made. As we told to reviewer 1 we are sorry for that but the Latin names were in italic in the manuscript we had submitted and this is probably a problem with the website!
Line 115 (revised version): “To this end the 2,2-Diphenyl-1-picrylhydrazyl (DPPH) assay was performed”
Line 491.: Fish Embryo Acute Toxicity (FET) Test is the reference 22 [22].
Line 1029-30 (revised version): (guidelines 236: Fish Embryo Acute Toxicity (FET) Test) [24]